# A R-Script for Generating Multiple Sclerosis Lesion Pattern Discrimination Plots

**DOI:** 10.3390/brainsci11010090

**Published:** 2021-01-12

**Authors:** Robert Marschallinger, Carmen Tur, Hannes Marschallinger, Johann Sellner

**Affiliations:** 1Department of Geoinformatics, University of Salzburg, Schillerstr 30, 5020 Salzburg, Austria; 2Department of Neurology, Christian Doppler Medical Center, Paracelsus Medical University, Ignaz-Harrer Str. 79, 5020 Salzburg, Austria; j.sellner@salk.at; 3Department of Neuroinflammation, Queen Square Multiple Sclerosis Centre, UCL Institute of Neurology, University College London, London WC1N 3BG, UK; c.tur@ucl.ac.uk; 4Neurology/Neuroimmunology Department, Multiple Sclerosis Centre of Catalonia, Vall d’Hebron University Hospital, 08035 Barcelona, Spain; 5Marschallinger GeoInformatik, Fischtagging 87, 5201 Seekirchen, Austria; hannes@marschallinger.eu; 6Department of Neurology, Landesklinikum Mistelbach-Gänserndorf, Liechtensteinstr. 67, 2130 Mistelbach, Austria; 7Department of Neurology, Klinikum Rechts der Isar, Technische Universität München, Ismaninger Str. 22, 81675 München, Germany

**Keywords:** multiple sclerosis, MS-lesion, MRI, geostatistics, R statistical computing

## Abstract

One significant characteristic of Multiple Sclerosis (MS), a chronic inflammatory demyelinating disease of the central nervous system, is the evolution of highly variable patterns of white matter lesions. Based on geostatistical metrics, the MS-Lesion Pattern Discrimination Plot reduces complex three- and four-dimensional configurations of MS-White Matter Lesions to a well-arranged and standardized two-dimensional plot that facilitates follow-up, cross-sectional and medication impact analysis. Here, we present a script that generates the MS-Lesion Pattern Discrimination Plot, using the widespread statistical computing environment R. Input data to the script are Nifti-1 or Analyze-7.5 files with individual MS-White Matter Lesion masks in Montreal Normal Brain geometry. The MS-Lesion Pattern Discrimination Plot, variogram plots and associated fitting statistics are output to the R console and exported to standard graphics and text files. Besides reviewing relevant geostatistical basics and commenting on implementation details for smooth customization and extension, the paper guides through generating MS-Lesion Pattern Discrimination Plots using publicly available synthetic MS-Lesion patterns. The paper is accompanied by the R script *LDPgenerator.r*, a small sample data set and associated graphics for comparison.

## 1. Introduction

Multiple sclerosis (MS), an inflammatory demyelinating disease of the central nervous system with neurodegenerative processes in the later course, affects more than 2.5 million people worldwide. It is the leading nontraumatic cause of serious neurologic disability in young adults. MS is initially characterized by phases of clinical relapses and remissions in 80–90% of the patients, and frequently followed by progression of disability over time. MS is highly variable—from benign to disastrous [1]: some patients may accumulate severe and irreversible disability within a few years, while others may show a benign course with just little or no disability even after decades. The hallmark of MS are lesions in the white and grey matter of the central nervous system, which are hyperintense on T2-weighted MRI sequences. MRI is the key technology to assess MS-lesion dissemination in space and time [2]. The number of MS-lesions, total lesion volume, spatial lesion pattern and single lesion shape are highly variable across patients with MS [3]. This makes the correlation of radiological data and clinical findings extremely challenging—a situation also known as the “clinicoradiological paradox” [4].

Variography, a core method of classical geostatistics, proved suitable for explorative data analysis (EDA) of MS-White Matter Lesion patterns (MS-WML) and for extracting quantitative spatial-statistics metrics on MS-WML. The MS-Lesion Pattern Discrimination Plot (MS-LDP) summarizes these metrics in a clear and standardized form, to aid in follow-up, cross-sectional and medication impact analysis [5,6]. With the aid of the MS-LDP, significantly different evolution of MS-lesion patterns could be disclosed between male and female early MS cohorts [7].

## 2. Materials and Methods

### 2.1. Sample Data

Three phantoms of brains with MS-lesions (MNI_mild, MNI_moderate, MNI_severe) that were used for both illustrating this paper and as the accompanying sample data were downloaded from: https://brainweb.bic.mni.mcgill.ca/brainweb/anatomic_ms.html.

### 2.2. Software

The R-script *LDPgenerator.r* (Appendix A, see Appendix B for description) is based on the statistical computing environment R [8] and was developed and tested in R version 3.6.0. *LDPgenerator.r*, the above-mentioned sample data and result files can be found in Appendix B. RNifti, a necessary R package can be downloaded and installed from the CRAN package repository: https://CRAN.R-project.org/package=RNifti.

### 2.3. Basics of the MS-Lesion Pattern Discrimination Plot

Below, the rationale behind and the making of the MS-Lesion Pattern Discrimination Plot (MS-LDP) is reviewed in compact form. For a more in-depth discussion of the clinical background, especially the application of the MS-LDP to real-world data sets, see [5,6,7]. As an example of MS-LDP graphics depicting a larger cohort of patients with MS that was processed with *LDPgenerator.r*, see *LDP_Supplement.jpg* in Appendix B (from [7], where also an interpretation of this MS-LDP can be found).

Based on geostatistical methods, *LDPgenerator.r* produces MS-LDP from either binary MS-WML masks or MS-WML probability maps in Montreal Normal Brain (MNB) geometry (Figure 1).

In the current context, a binary MS-WML mask is a voxel array that represents MS-lesions with voxel value of 1 and all other voxels with value of 0. Such masks are derived from MRI data either by expert manual labelling the MS-lesions with specific software (e.g., MRIcron, [9]) or by automatic MS-lesion extraction software (e.g., LST, [10]). Automatic MS-lesion extraction software yields MS-WML probability maps with lesion probability values between 0.0 and 1.0. These can be internally binarized by *LDPgenerator.r*, employing a user-supplied threshold. MNB geometry means that individual brain geometry is mapped to the Montreal Normal Brain template [11], usually with the help of dedicated software (e.g., SPM [12] or FSL [13]).

Geostatistics comprises a range of algorithms for characterizing, modelling [14] and simulation [15] of multidimensional data and associated uncertainty, including spatiotemporal data [16]. Originally developed for mining optimization [17], geostatistics subsequently found the way into a variety of space-time related fields. In medicine, early applications of geostatistics deal with the creation of geomedical maps, showing the geographical distribution of disease cases (https://www.esri.com/library/ebooks/geomedicine.pdf) [18]. Possible applications of geostatistics go far beyond 2D mapping in the macro scales, however. For example, geostatistical methods can be readily employed in the micro scales [19,20] and enable quantitative texture analysis [21,22]. Since this paper focuses on the application of the variogram to texture analysis of MS-lesion patterns, a short excursus to the foundation of variography seems reasonable. The variogram is a central EDA tool in classical two-point geostatistics (“classical” as compared to the more recent approach of Multiple-Point Geostatistics [23]). The variogram (Figure 2) quantifies the spatial structure of measurements by contrasting the distance between pairs of measurement points (abscissa) and the associated variability (*γ*, on the ordinate). In other words, the variogram yields a measure of spatial correlation [24,25].

The empirical variogram *γ*(h) is straightforwardly computed using (Equation (1)): (1)$$\gamma \left(h\right)=\frac{1}{2n\left(h\right)}*\overset{n}{\underset{i=1}{\sum }}({\left(z\left(x_{i}\right)-z\left(x_{i}+h\right)\right)}^{2}$$
z(x) value of variable at some 3D location x, here z(x) is a voxel with z = binary variable (0 or 1); h lag vector of separation between observed data (units: here, mm); n(h) number of data pairs [z(x), z(x+h)] at lag h; *γ*(h) empirical variogram value for lag h.

The *γ*(h) of a binary MS-WML is calculated by comparing the values (0 or 1) of all voxel pairs within a specified lag h, according to Equation (1). Calculating *γ*(h) for increasing lag distances h, the empirical variogram plot (short: “the variogram”) is derived (Figure 2). 

Variograms of binary MS-WML generally start with small values of *γ* at small h (distance in Figure 2), which is due to the large correlation of adjacent voxel pairs. After an increase in *γ* with lag away from the origin, with further increases in h the correlation decreases, and the variogram levels off. The flatter the variogram near its origin, the more pronounced is the spatial correlation (i.e., in the current context: the larger the MS-lesions will be, on average). Computing variograms for specific lag orientations yields so-called directional variograms. Individual directional empirical variograms in the three major orthogonal directions of MNB geometry (x, y, z directions, compare Figure 1) can be used to disclose and quantify spatial anisotropies of MS-lesion patterns. 

Since empirical variograms focus on visual inspection, several permissible variogram functions for quantifying empirical variograms were introduced [26]. These functions approximate an empirical variogram’s shape by two parameters: the variogram range *a*, and the variogram sill *c*. Among the available variogram model functions, the exponential variogram model was found to be the most suitable for quantifying MS-lesion patterns [5]: (2)$$\gamma \left(h\right)=c\left(1-\exp\left(\frac{-3\left|h\right|}{a}\right)\right)$$
*c* Sill; *a* Range; h lag vector of separation; *γ*(h) model variogram value for lag h.

When exponential variogram models are separately fitted to x, y, z directional variograms (Equation (2)), three value pairs are derived: a[X],c[X]; a[Y],c[Y]; a[Z],c[Z]; (with a[X],c[X]; … values of a, c in direction x, etc.). Plotting the natural logarithm of above three value pairs, the Component MS-LDP is produced (Figure 3a). It yields information on MS-WML geometrical anisotropies—e.g., lesion confluence along the CSF system, isotropic growth of individual lesions or Dawson fingers. The MS-LDP (Figure 3b) abstracts MS-WML geometry with just two parameters in combining above three value pairs by their means (Equation (3)):(3)$$\bar{a}=\ln(mean\left(a\left[X\right],a\left[Y\right],a\left[Z\right]\right),\;\bar{c}=\ln(mean\left(c\left[X\right],c\left[Y\right],c\left[Z\right]\right)$$

The MS-LDP quantifies important geometric aspects of MS-WML: *a* is considered a measure of spatial continuity, *c* is proxy of total lesion load. The higher *a*, the bigger and smoother (i.e., with less lesion surface roughness) are lesions; the higher *c*, the higher is total MS-lesion load. From analyzing a larger cohort of consistently acquired MS-WML, abscissa *a*, and ordinate *c* of the MS-LDP were defined to span: *a* … 0 to 3, *c* … −12 to −4 [6]. Figure 3a is a Component MS-LDP and Figure 3b is the associated MS-LDP, produced from MNI_mild, MNI_moderate and MNI_severe data.

## 3. Results (Developed Code)

### 3.1. LDPgenerator.r: Program and Data Flow

Figure 4 shows *LDPgenerator.r* code sections and sequence of operations. In short, the program flow is as follows: after the user has selected input and output files, for each input file, empirical variograms are calculated and exponential variogram models are fitted. Variogram graphics are generated and Component MS-LDP and MS-LDP parameters are stored in a container file. After processing all input files, plots are generated from the container file. 

Below, code sections 1–5 are described in more detail (see Figure 4 and compare respective comments in *LDPgenerator.r* code):
Section 1 (code lines 11–12):

Package RNifti is loaded; this package is prerequisite for fast access to images stored in Nifti-1 or Analyze-7.5 medical image formats. 

Section 2 (code lines 14–29):

Default values for variography parameters can be set by the user: image (binarization) threshold in case MS-lesion probability maps from automatic MS-lesion segmentation programs are to be processed, nonlinear least squares (nls) starting estimates, number of lags, graphics appearance. 

*ImageThreshold*: The standard input to LDPgenerator is binary MS-WML, i.e., a binarized voxel array with MS-lesion voxels = 1 and all other voxels = 0. When floating-point MS-lesion probability maps with MS-lesion probabilities between 0.0 and 1.0 are input, ImageThreshold controls the conversion to binary: all voxels with values above ImageThreshold are set to 1.0 (MS-lesion), voxels with lower values are set to 0.0 (non-lesion). 

*Max_lag*: Defines the number of distance classes (lags) used in variography. Each distance class is one voxel wide, i.e. measured in voxel dimension. Variograms of binary MS-WML should be confined to distances of 0–15 mm, because this range holds the most relevant correlation information [5]; i.e., if voxel dimension is 1.5 mm, Max_lag should be set to 10.

*Guess_a, Guess_C*: nls starting estimates for the Exponential Variogram Model. For further information on nls (Nonlinear Least Squares function) parameters, see the R documentation.

*LDPxdim, LDPydim*: Size of LDP and Variogram graphics output, in pixels (width, height).

*LDPsymbolsize, LDPtextsize*: Relative size of symbols and annotation text in MS-LDP graphics.

*VarioGraphicsPostfix*: File postfix of Variogram graphics per Nifti/Analyze file.

Section 3 (code lines 32–36): Input and output files are selected; LDPgenerator.r uses the straightforward graphical user interface (GUI) of base R.

*List of input files (*.nii, *.img, *.hdr):* Input files can be selected from the list presented by the GUI. Input files must be formatted Nifti-1.0 (file type *.nii) or Analyze-7.5 (file pairs of type **.hdr,*.img*). Mixing Nifti and Analyze files is possible. Analyze files can be selected by selecting either .*hdr* or *.img*. Make sure input files are explicitly 3D in the image headers (!)

*LDP container file (*.var)*: The user provides a file name, with standard extension *var*. The LDP container file (ASCII) contains variography data (compare Table 1) on each input file to generate the Component MS-LDP and MS-LDP graphics. 

*LDP graphics file (*.png):* The user provides a file name, with standard extension .png. After processing all input files, LDPgenerator.r produces the MS-LDP (Figure 3b). 

*LDP component graphics file (*.png):* The user provides a file name, with standard extension .png.

After processing all input files, LDPgenerator.r produces the Component MS-LDP with a, c symbols for x, y, z directions (compare Figure 3a). Graphics file format is portable network graphics.

Section 4 (code lines 46–207): In the central processing loop, for each input file, the following operations are performed:Section 4a (code lines 51–69): RNifti extracts a voxel array and associated geometry data (number of voxels in x, y, z direction, xyz dimension per voxel).Section 4b (code lines 71–76): The extracted voxel array is thresholded, yielding a classified result (1 = MS-lesion, 0 = rest).Section 4c (code lines 78–139): Three individual empirical variograms are calculated—one per voxel array x, y, z direction, for lags 1 to Max_lag. Lag values and pair counts are stored.Section 4d (code lines 141–175): An exponential variogram model is separately fit to each x, y, z empirical variograms, using the R nls function with starting estimates Guess_a, Guess_C. Derived model parameters ax, Cx, ay, Cy, az, Cz, and mean a, mean c values are stored for x, y, z directions.Section 4e (code lines 177–200): Per input file, empirical variogram graphs and associated exponential variogram model functions for individual x, y, z directions are displayed and stored in png format, with associated filenames.Section 4f (code lines 202–207): The index number (1 ... n) of processed MS-WML, the natural logarithm of model parameters mean(ax, ay, az), mean(Cx, Cy, Cz) and directional components ax, Cx, ay, Cy, az, Cz, and the respective input file name are appended to the LDP container file (ASCII). File contents are a good starting point for postprocessing geostatistical data on MS-WML (Table 1). MS-WML index numbers are displayed in MS-LDP and Component MS-LDP graphics to reference input file names while not overloading graphics.Section 5 (code lines 209–248): MS-LDP and Component MS-LDP graphics are displayed and exported in png format. LDPgenerator.r terminates.

### 3.2. A Worked Example in 5 Steps

The associated sample data set comprises three MS-lesion probability maps (“brain phantoms”) downloaded from brainweb https://brainweb.bic.mni.mcgill.ca/brainweb/anatomic_ms.html, converted to the Analyze format. First, copy the file pairs MNI_mild.hdr, MNI_mild.img; MNI_moderate.hdr, MNI_moderate.img; MNI_severe.hdr, MNI_severe.img (files included in Appendix A, see Appendix B for description) to a suitable directory. Then follow the 5 steps below:
Step 1: From standard R, launch LDPgenerator.r (no changes of parameters necessary in the script);Step 2: Define input files: Navigate to the relevant directory and select MNI_mild.hdr, MNI_moderate.hdr and MNI_severe.hdr from the file list;Step 3: Define the LDP container output file: In the highlighted input box, type MNI.var (Appendix A, see Appendix B for description);Step 4: Define the LDP graphics output file: In the highlighted input box, type MNI_LDP.png (Appendix A, see Appendix B for description);Step 5: Define the Component LDP graphics output file: In the highlighted input box, type MNI_LDP_xyz.png (Appendix A, see Appendix B for description).

After step 5, *LDPgenerator.r* sequentially opens the selected input files and creates associated graphics containing variograms, variogram models and model parameters in the input file directory: *MNI_mild.hdr_variograms.png, MNI_moderate.hdr_variograms.png, MNI_severe.hdr_variograms.png* (files included in Appendix A, see Appendix B for description). Moreover, variography parameters are appended to the LDP container file *MNI.var*. To avoid overloading graphics with labels, in *MNI.var* the input files are sequentially numbered from 1 … n, graphical elements in *MNI_LDP_xyz.png* and *MNI_LDP.png* are labelled accordingly. 

Table 1 shows contents of LDP container file (ASCII) with variogram model parameters: ID, variogram model parameters and associated MS_WML file names.

As further example, a “real-world” MS-LDP that was used in a clinical study is included in Appendix A (*LDP_Supplement.jpg*, Appendix A, see Appendix B for description). See [7] for details and in-depth interpretation.

## 4. Discussion

Based on the widespread, freely available R statistics environment, *LDPgenerator.r* provides routine production of MS-Lesion Pattern Discrimination Plots, associated variogram graphics and statistics.

The first implementation for generating MS-LDP that was used in [5,6,7] involved operating-system based scripting and incorporated, besides R scripts, two MS-Windows based software components. Obviously, this was tedious, error-prone and limited the generation of MS-LDP to the MS-Windows operating system. As compared to above original implementation, MS-LDP can now be produced easily in standard R, by just selecting input files and defining output files via the R standard graphical user interface.

The script runs with acceptable speed, processing time for a typical MS-WML is about one minute on a current PC. Clearly, an interpreted language like R has drawbacks regarding processing speed. There are several opportunities for script improvement, however: *LDPgenerator.r* involves several loops, parts of which could be re-formulated as optimized user defined functions, which in turn would speed up the script. Except of package RNifti, the current version of *LDPgenerator.r* is implemented on base R. Since the R environment enables straightforward code extension with a wealth of packages, here are some ideas for customization/improvements:
Adding a 3D viewer: The recent version of RNifti (1.1) contains a basic viewer for easy implementation of Nifti/Analyze viewing. Providing interactive viewing of MS-WML would facilitate setting the correct binarization threshold values, via enabling visual checking of resulting lesion probability maps.Using a GUI package, e.g., https://r4stats.com/articles/software-reviews/r-gui-comparison/.Availability of state-of-art GUI elements like customizable buttons, input boxes or spinners would enable easy tuning of variography and graphics parameters.Using an improved graphics package—see: https://cran.r-project.org/web/views/Graphics.html. More sophisticated annotation elements like labels, scalable arrows, or improved legend elements would facilitate explorative data analysis of time series portrayed in the MS-LDP, e.g., data from follow-up MRI.

*LDPgenerator.r* was developed with R version 3.6.0 and tested with input data from variable sources. Some pitfalls occurred that need to be mentioned: *LDPgenerator.r* can process only Nifti and Analyze files that are declared “3D” in their file headers. Using R Studio to launch *LDPgenerator.r*, graphics generation failed with high-resolution (4k) graphics hardware. *LDPgenerator.r* was tested under Windows 7 and Windows 10 operating systems. While no problems occurred under Windows 7, under Windows 10, a *Qt: Untested Windows version 10.0 detected!* message is issued. In the current context this warning can be ignored, however.

## Figures and Tables

**Figure 1 brainsci-11-00090-f001:**
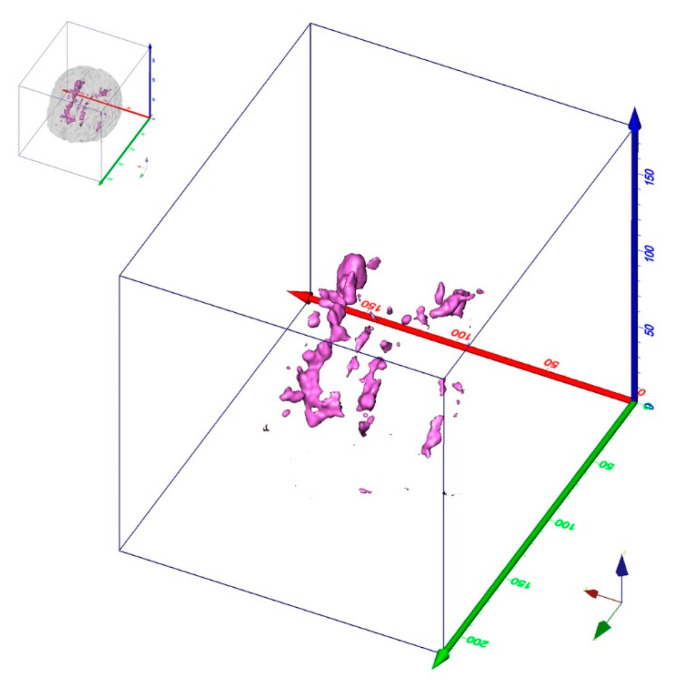
3D-view of binary MS-lesion mask in MNB geometry (phantom of human brain with severe MS lesions, binarized (threshold value = 0.5). MS-lesions (with voxel value of 1) are magenta, all other volume (with voxel value of 0) is transparent. Small insert top left indicates position of MS-lesions within white matter (transparent grey). Axis colors: x = red, y = green, z = blue. Axis scale: millimeters.

**Figure 2 brainsci-11-00090-f002:**
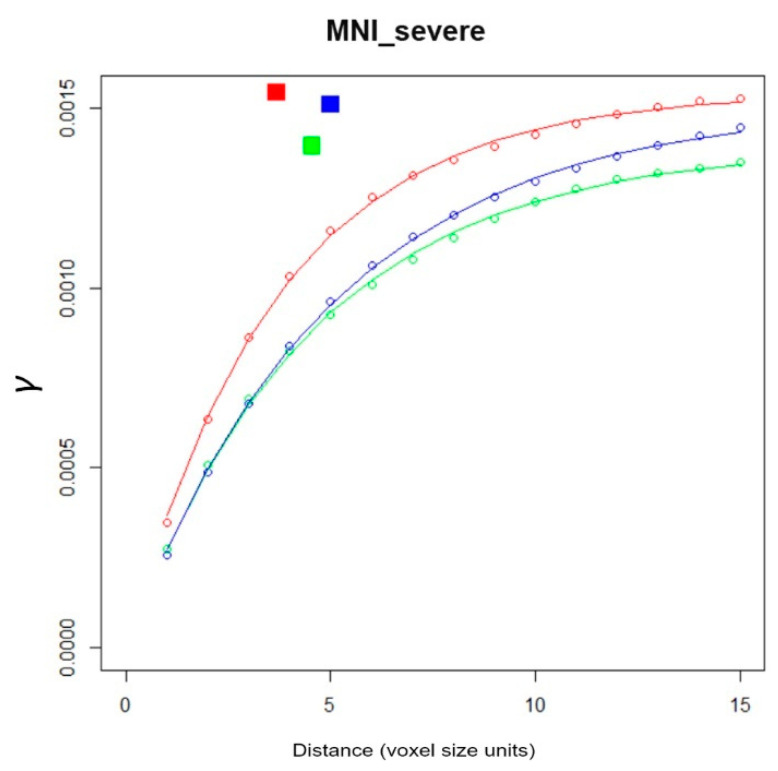
Variogram plot: Distance (abscissa) vs. *γ* (ordinate) with directional empirical variograms, separately fitted variogram models and associated model parameters *a*, *c*. Red, green, blue colors refer to variography x, y, z directions. Dots represent empirical directional variogram values, lines are fitted exponential variogram models, parameters *a*, *c* define coordinates of squares as derived from variogram modeling. Produced with *LDPgenerator.r* from binary MS-WML mask in Figure 1. See text for details.

**Figure 3 brainsci-11-00090-f003:**
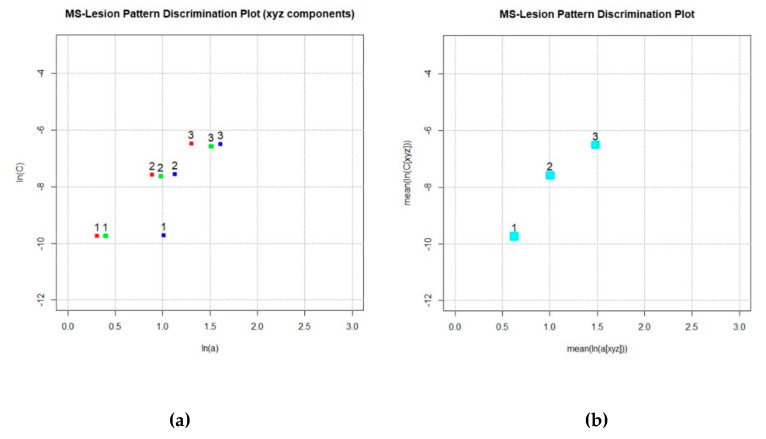
(**a**) Component MS-LDP, ln(*a*[x,y,z]) vs. ln(*c*[x,y,z]) for MS-WML: MNI_mild (label = 1), MNI_moderate (label = 2) and MNI_severe (label = 3). Each MS-WML geometry is expressed by x, y, z directional variogram model parameters, indicated by red, green, blue squares. For MNI_mild, a strong anisotropy according to MS-lesions stretched in z direction is indicated by relatively higher ln(*a*[z]). (**b**) MS-LDP, mean (ln(*a*[xyz]) vs. mean (ln(*c*[xyz]) for MNI_mild (label = 1), MNI_moderate (label = 2) and MNI_severe (label = 3). MS-lesion pattern (1, 2, 3) geometry is abstracted to one point each. This is useful to avoid overloading graphics when working with a larger number of MS-WML (e.g., [5,6], compare the real-world MS_LDP: *LDP_Supplement.jpg*, in Appendix B).

**Figure 4 brainsci-11-00090-f004:**
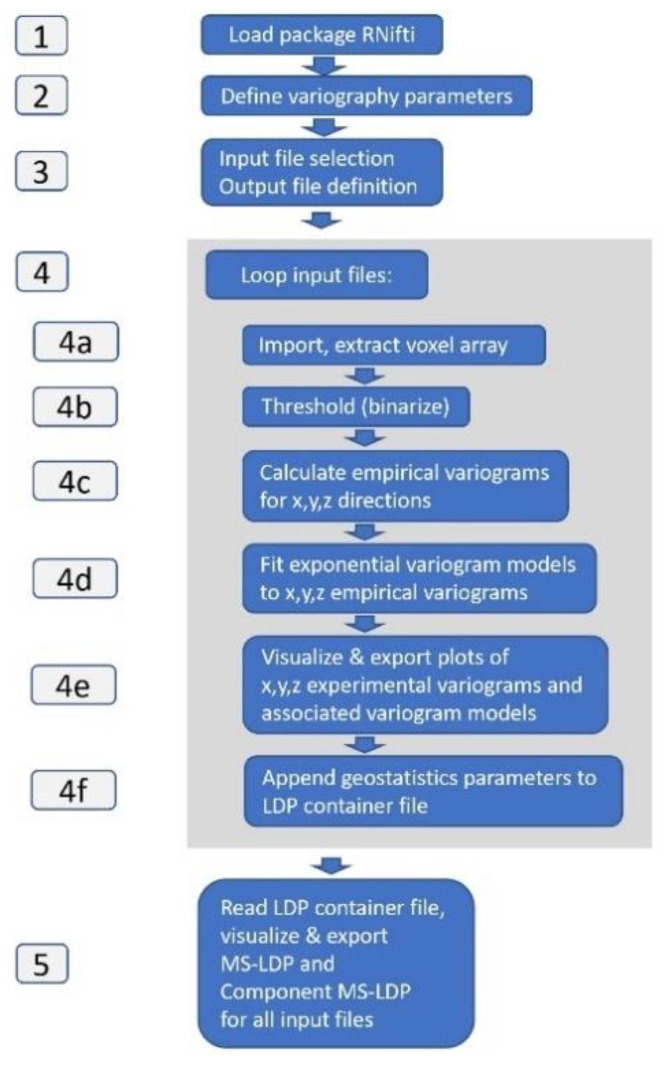
*LDPgenerator.r* program and data flow chart, indicating sequence of operations (blue labels) involved in generating MS-LDP. Grey labels are code sections, central processing loop is on grey background.

**Table 1 brainsci-11-00090-t001:** Contents of variogram parameter file MNI.var.

ID	ln(avg(a[xyz]))	ln(avg(C[xyz]))	ln(aX)	ln(CX)	ln(aY)	ln(CY)	ln(az)	ln(CZ)	File
1	0.62610	−9.74182	0.30975	−9.74533	0.40066	−9.75081	1.01342	−9.72945	MNI_mild.img
2	1.00625	−7.59234	0.89172	−7.58171	0.98117	−7.63554	1.13109	−7.56123	MNI_moderate.img
3	1.48463	−6.51406	1.30517	−6.47390	1.51387	−6.57510	1.61090	−6.49598	MNI_severe.img

## Data Availability

The test data presented in this study are available in the Appendix A.

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
