# Peer review of "A R-Script for Generating Multiple Sclerosis Lesion Pattern Discrimination Plots"

_brainsci, 2021, doi:10.3390/brainsci11010090_

Round 1
Reviewer 1 Report
Disclaimer: I'm by no means an expert in the area, but rather read this from a stats-savvy but otherwise layman's perspective. I learned a couple of things upon reading, so that is good. Some comments that might render this accessible to a broader audience, though I would leave it to the authors to address it or not - if things are too obvious for the intended audience, there is no point in explaining them further, while other things need to be corrected and/or improved to maximize impact.
* 70f / Figure 1 caption: I do not get the meaning of voxel value == 0 or 1. To me, a voxel is a point in space (actually a cube), but it is unclear to me what the respective value implies (other than 1 being mangenta and 0 transparent). Possibly worth a half-phrase to clarify. It becomes a little clearer in the sequence, but would help to clarify here.
* 78: I don't think that the majority of references 14-20 is really pertinent to this paper, and I do not think it helps much to provide an seemingly random selection of papers on geostatitics. I think it would be better to limit this to a textbook or two that clearly cover the aspects of relevance here; any other literature should either be referred to individually in the text (if there is a relevant link) or dropped. It doesn't help the reader to have go through a bunch of references w/o any guidance, and find out that some are irrelevant.
* 80-84: this mathematical description is likely unaccessible to many - I suggest a few lines to explain what it does heuristically such that it appeals also to non-statisticians, say. What's the motivation of the variogram, say? What does gamma(h) express? Same in the sequence: a lot of this is based on a purely technical description, but is lacks a technically low-level motivation/description of what it actually does, e.g. the variogram plot in fig 2, or eqn (2)
* 81/84: To me, the formula clearly has a "y(h)", while in the sequence it is rather "\gamma(h)" Might be a matter of fonts, though I doubt it. Needs to be corrected unless I am missing sth. Same in 107.
* 95ff: I don't think it can be written this way: "after an initial increase..." suggests that there is no increase anymore later, but there is in the entire Fig 2. You probably rather mean a *steep* increase or alike, but as is, it doesn't seem to make sense to me
* 129: remove extra full stop
* I don't think section 3 is "results" - it is rather the tool developed here; some of section 2 seems more like "results" than most of section 3. Might require a little re-organization to get that straight.
* Fig 4: looks like the picture was copied ouf of word or PPT with spell checker (also in the wrong language) on - update the Figure to remove all of the red lines
* 203: I suspect starting with "3.2" should be a new subsection header, otherise the entire line reads very odd
* 206ff: There are many more options to download from that website, so it is unclear to me where the files really are. You are likely rather referring to the appendix to this ms, which should be made clearer.
* 241: "in" rather than "on"
* 261: In the submission files, the ZIP file has a different name. Make sure it matches ultimately.
* 332f: ???
Reviewer 2 Report
In this paper, the authors propose an R script that generates multiple sclerosis Lesion Pattern Discrimination Plots from either Nifti-1 or Analyze-7.5 format files. More in detail, the code provided computes an empirical variograms for each dimension, and then provides an exponential fitting of the variograms. The MS-Lesion Pattern Discrimination Plots consist in a scatter plot of the two parameters of the model.
The paper is generally well written and clear, with only minor mistakes (e.g. “proofed” instead of “proved” in line 41, “Montral” instead of “Montreal” in line 62). The description of the operations performed by the proposed code is properly explained and the utility of the MS-Lesion Pattern Discrimination Plot is described, albeit succinctly. There appears to be some problems with the terminology used in eq. 1, as the same value seems to be indicated with y in the equation and with γ in the text. The expression ¦h¦ is furthermore used, whose meaning I am unfamiliar with.
This work is not a research paper, per se, as no experiments are proposed or conducted. Instead, the script presented is tested on three phantom data sets. A test on actual data to demonstrate the classification power of the obtained MS-LPDPs might have increase the appeal of the work.
